# Role of Different Members of the *AGPAT* Gene Family in Milk Fat Synthesis in *Bubalus bubalis*

**DOI:** 10.3390/genes14112072

**Published:** 2023-11-13

**Authors:** Zhipeng Li, Ruijia Li, Honghe Ren, Chaobin Qin, Jie Su, Xinhui Song, Shuwan Wang, Qingyou Liu, Yang Liu, Kuiqing Cui

**Affiliations:** 1Guangxi Key Laboratory of Animal Reproduction, Breeding and Disease Control, State Key Laboratory for Conservation and Utilization of Subtropical Agro-Bioresources, Guangxi University, Nanning 530004, China; 2018302017@st.gxu.edu.cn (R.L.); 2018301027@st.gxu.edu.cn (H.R.); 2018401005@st.gxu.edu.cn (C.Q.); 18438591697@163.com (J.S.); songxinhui@st.gxu.edu.cn (X.S.); wswan@st.gxu.edu.cn (S.W.); 2Guangdong Provincial Key Laboratory of Animal Molecular Design and Precise Breeding, School of Life Science and Engineering, Foshan University, Foshan 528225, China; qyliu-gene@fosu.edu.cn (Q.L.); kqcui@fosu.edu.cn (K.C.); 3Guangxi Zhuang Autonomous Region Center for Analysis and Test Research, Nanning 530022, China; gxatly@sina.com

**Keywords:** buffalo mammary epithelial cells, cell transfection, gene expression, RNA interference, triacylglycerol compositions

## Abstract

During triacylglycerol synthesis, the acylglycerol-3-phosphate acyltransferase (AGPAT) family catalyzes the conversion of lysophosphatidic acid to phosphatidic acid and the acylation of sn-2 fatty acids. However, the catalytic activity of different AGPAT members is different. Therefore, this study aimed to investigate the mechanism through which different *AGPAT*s affect the efficiency of TAG synthesis and fatty acid composition. The conservation of amino acid sequences and protein domains of the AGPAT family was analyzed, and the functions of *AGPAT1*, *AGPAT3*, and *AGPAT4* genes in buffalo mammary epithelial cells (BMECs) were studied using RNA interference and gene overexpression. Prediction of the protein tertiary structure of the AGPAT family demonstrated that four conservative motifs (motif1, motif2, motif3, and motif6) formed a hydrophobic pocket in AGPAT proteins, except AGPAT6. According to cytological studies, *AGPAT1*, *AGPAT3*, and *AGPAT4* were found to promote the synthesis and fatty acid compositions of triacylglycerol, especially UFA compositions of triacylglycerol, by regulating *ACSL1*, *FASN*, *GPAM*, *DGAT2*, and *PPARG* gene expression. This study provides new insights into the role of different *AGPAT* gene family members involved in TAG synthesis, and a reference for improving the fatty acid composition of milk.

## 1. Introduction

Buffaloes produce the second-highest quantity of milk in the world, and buffalo milk contains higher contents of milk fat, milk protein, and total solids than milk from other livestock [1,2,3]. One of our previous studies showed that the milk fat in buffalo is significantly higher than that in the milk of the Holstein breed (7.88 ± 0.91 vs. 4.24 ± 0.80) [1]. The milk fat content and composition are crucial factors affecting milk flavor, and nutritional and economic value. Milk fat is among the main fat sources for humans, especially for Westerners. A high-fat diet, particularly the intake of saturated fatty acids (SFAs), has recently been considered the most crucial factor leading to hyperlipidemia and other cardiovascular diseases [4]. However, compared to Holstein milk, buffalo milk is rich in unsaturated fatty acids (UFAs), such as linoleic, linolenic, conjugated linoleic acid, eicosapentaenoic acid, and arachidonic acids, which are considered beneficial to human health [5]. Therefore, increasing the unsaturated fat content in milk has become a critical direction in dairy animal breeding. However, information available regarding the mechanism underlying UFA biosynthesis in buffalo milk is inadequate.

Fatty acids in milk are mainly taken up from blood (plasma) during the first lactation month, or derived from de novo synthesis by mammary cells from the second lactation month [6]. Triacylglycerol (TAG), which comprises 98% of the fat in buffalo milk [7], is synthesized from fatty acids and glycerin by enzymes, including glycerol-3-phosphate acyltransferases (GPATs), acylglycerol-3-phosphate acyltransferases (AGPATs), phosphatidic acid phosphatase, and diacylglycerol acyltransferases (DGATs) [8]. Among these, AGPATs catalyze the conversion of lysophosphatidic acid (LPA) to phosphatidic acid (PA), in addition to further dephosphorylation to form diacylglycerol (DAG). *AGPAT1* and *AGPAT6* are highly expressed in both buffalo and cattle mammary glands during lactation, and *AGPAT1* or *AGPAT6* knockdown can significantly decrease the TAG content in mammary epithelial cells by regulating the expressions of lipogenic-related genes [9]. Moreover, GPATs prefer to catalyze SFA acylation, whereas AGPATs prefer to catalyze UFAs [8,10]. During TAG synthesis, the AGPAT family catalyzes the acylation of sn-2 fatty acids. Meanwhile, the sn-2 site of TAGs is mostly UFA, which further indicates that AGPAT may have a higher affinity for UFAs. Another study showed that different AGPAT family members have different affinities for fatty acid acyl-CoA, among which AGPAT3 and AGPAT4 exhibited a significant preference for the polyunsaturated fatty acid (PUFA) acyl-CoA [11,12]. Therefore, systematically revealing the effects of different *AGPAT* gene family members on milk fat synthesis to increase the UFA content of milk is of great significance.

In this study, we aimed to investigate the mechanism through which different AGPATs affect the efficiency of TAG synthesis and fatty acid composition. We analyzed the conservation of amino acid sequences and protein domains of the AGPAT family, and studied the function of the *AGPAT1*, *AGPAT3*, and *AGPAT4* genes in buffalo mammary epithelial cells (BMECs) via RNA interference and gene overexpression. Our study may provide novel insights into the function of the *AGPAT* gene family and offer references for the molecular breeding of dairy cows.

## 2. Materials and Methods

### 2.1. Experimental Animals and Sampling

The buffalo mammary gland tissues used in this study were collected in the Buffalo Breeding Farm of the Buffalo Research Institute, Chinese Academy of Agricultural Sciences, Nanning, Guangxi, China. Buffalo milk was also collected during early (30–100 days), mid (100–200 days), and late (>200 days) lactation using the aforementioned methods [13]. All of the selected buffaloes were in the second or third parity, with ages between 6.5 and 7 years. In brief, the milk samples were collected in summer (June–July), between 5:00–6:00 a.m., on the same day for each lactation stage. The samples were collected manually into sterile RNase-free tubes, taking care to avoid any contamination. The samples were immediately placed on ice and transported to the laboratory, and stored at −80 °C until further use.

### 2.2. Extraction of Total RNA from Milk Fat Globules

RNA from milk fat globules (MFG) was extracted and used for gene expression analysis. By referring to a previous report [1], the total RNA was extracted from the MFG at 4 °C and completed within 2 h to improve the quality of the mRNA. In brief, the milk samples were centrifuged at 2000× *g* for 10 min at 4 °C to isolate milk fat. The supernatant fat layer was separated, and 500 μL of fat was mixed with 1 mL of TRIzol solution (Invitrogen, Carlsbad, CA, USA). Following centrifugation (4000 rpm, 5 min), the top layer of fat was removed, and the bottom liquid was separated and mixed with 200 μL of chloroform. RNA from BMECs was also extracted using TRIzol solution (Invitrogen, Carlsbad, CA, USA). The RNA was precipitated using isopropyl alcohol and dissolved with 30 μL of RNase-free water. Agarose gel electrophoresis was performed, and only mRNA samples with a low 5S band were selected for the following experiment.

### 2.3. Single-Strand cDNA Synthesis

The RNA purity was evaluated based on absorbance readings (ratio of A260/A230 and A260/A280) using a Nano-Drop ND-2000 spectrophotometer (Thermo Fisher Scientific, Waltham, MA, USA). The genomic DNA was removed using DNase treatment. First-strand cDNA was then synthesized using the RevertAid First Strand cDNA synthesis kit (K1622, Thermo Fisher Scientific, Waltham, MA, USA). The single-strand cDNA obtained was stored at −20 °C.

### 2.4. Phylogenetic, Secondary Structure, and Multiple Sequence Alignment Analyses

Amino acid sequences of the AGPAT proteins of buffalo, cow, goat, sheep, camel, human, and mouse were downloaded from the NCBI database (Appendix A) and were aligned with ClustalW using MEGA11 software (v11.0.11) [14]. A single alignment file was prepared. A neighbor-joining phylogenetic tree with a bootstrap value of 1000 replicas was constructed in MEGA11 (v11.0.11) [14]. The phylogeny was further optimized using the website tool of iTOL [15]. Motif analysis was performed using the MEME suite tool (v6) [16]. Using the NCBI conserved domain database [17] and CD-Search tool (v3.20) [18], the conserved domains of the *AGPAT* gene family were analyzed. TB tools (v1.098745) were used to integrate the results of the tree, motif, and protein-conserved domain analyses [19]. The amino acid sequences of all of the AGPAT proteins of buffalo and cattle were submitted to the online server Phyre2 (v2.0) [20] for designing the three-dimensional structure of each AGPAT protein, and the secondary structure features fold recognition end homology modeling was identified. The models with confidence higher than 99% were selected for subsequent analysis. In the tertiary structure, the motifs were characterized using PyMOL software (v2.5.2) [21] and distinguished using different colors. The electrostatic potential energy of the proteins were also predicted using PyMOL software (v2.5.2) [21]. The domains of the proteins were further characterized using the InterPro database [22].

### 2.5. Isolation, Culture, and Purification of Buffalo Mammary Epithelial Cells

The buffalo mammary epithelial cells (BMECs) were cultured and purified as reported previously [23]. Briefly, fresh buffalo mammary gland tissue was obtained from the butchery and washed three times with normal saline (0.9% NaCl). The acinus portion was extracted from the mammary gland tissue, washed with normal saline (0.9% NaCl), and transferred into high-resistance PBS (containing 400 IU mL^−1^ each of penicillin and streptomycin) until it was brought back to the laboratory. Then, the tissue pieces were placed in culture dishes on a clean bench, cut into 1–2 mm pieces, tiled on the bottom of the culture dish, and cultured in an incubator (38.5 °C) for 4 h. The tissue pieces were then inverted and cultured with F12/DMEN (Gibco, Waltham, MA, USA) containing 20% serum (Gibco, Waltham, MA, USA) in the upright position overnight. Once the epithelial cells started growing after approximately eight days, they were isolated through trypsin digestion combined with a cell adherence speed method. The purification procedure was repeated three times, and BMECs at 3–4 generations in the subculture were used for the following studies. The cells were cultured with F12/DMEN (Gibco, Waltham, MA, USA) containing 20% serum (Gibco, Waltham, MA, USA) with a cell incubator (Thermo Fisher Scientific, Waltham, MA, USA).

### 2.6. siRNA Synthesis and Overexpression Vector Construction of Buffalo AGPAT1, AGPAT3, and AGPAT4

siRNAs targeting AGPAT1, AGPAT3, and AGPAT4 were designed and synthesized by Sangon Biotech (Shanghai, China) with a control sequence (Appendix A). The *AGPAT1*, *AGPAT3*, and *AGPAT4* genes were cloned from BMECs according to the GenBank [24] sequences and inserted into the pcDNA3.1-eGFP vector to construct the respective overexpression vectors. 

### 2.7. Transfection of BMECs

Transfection of siRNAs and overexpression vectors was performed using Lipofectamine™ 2000 (Invitrogen, Carlsbad, CA, USA) according to the manufacturer’s instructions. The overexpression vectors or siRNAs were added to the transfection reagent at a 1:2 ratio after the cell density reached 80%. At 24 h after the transfection, fluorescence was detected to determine the transfection efficiency. The cells were collected 48 h after the transfection, and qRT-PCR was performed to analyze the gene expression. Each transfection experiment was performed, and each sample was detected three times. Cells transfected with pcDNA3.1-eGFP vector, or random siRNA sequences were used as the negative control. 

### 2.8. Extraction and Component Analysis of Fatty Acids

Fatty acid extraction and gas chromatography analysis were performed following a previous study [1]. In brief, the BMECs transfected with siRNAs or overexpression vectors were collected in a 100 mL colorimetric tube. A solution containing 2 mL of 95% ethanol, 4 mL of water, and 10 mL of 8.3 mol/L hydrochloric acid was added to the tube. The sample was extracted 3 times using a mixture of petroleum ether and ether. The combined extract was transferred into a new flask. The extracted fat was dried, weighed, and finally dissolved in hexane. The fatty acid composition was determined through gas chromatography using a Shimadzu GC-2014C (Kyoto, Japan) gas chromatograph equipped with an FID and a capillary column (30 m × 0.32 mm i.d.; film thickness: 0.25 μm) (Agilent DB23, Loveland, CO, USA). The injection port was set at 230 °C and the detector at 280 °C. The column was maintained at 180 °C for 5 min and heated up to 230 °C at 3 °C min^−1^. The carrier gas was maintained in high-purity nitrogen, and the injection volume was 1 μL. Individual FA methyl esters were identified through a comparison with a standard mixture of 37 Component FAME Mix (Supelco Analytical Products, Bellefonte, PA, USA). The standards of PUFA-2, nonconjugated C18:2 isomer mixture, and cis-5,8,11,14,17 C20:5, cis-4,7,10,13,16,19 C22:6 (Supelco Analytical Products, Bellefonte, PA, USA), cis-6,9,12 C18:3, and cis-9,12,15 C18:3 (Matreya LLC, Pleasant Gap, PA, USA) were used to identify the PUFAs. The C18:1 isomer was identified based on commercial standard mixtures (Supelco Analytical Products, Bellefonte, PA, USA) and published isomeric profiles. Using a nonadecanoic acid as an internal standard, the veracity of peak normalization was increased. For all of the studied fatty acids, the coefficient of variation [(SD/mean) × 100] was <3.5%, which suggested good repeatability of the GC data. All of the samples were detected three times. All fatty acid compositions of BMECs were expressed as mg per 100 g of fat.

### 2.9. qRT-PCR Analysis

The primers (Appendix A) designed using Oligo 7.0 software [25] were synthesized by GenSys Biotech (Nanning, China). Using the SYBR qPCR master mix (Vazyme Biotech Co., Ltd., Nanjing, China) [26], qRT-PCR was performed following the manufacturer’s instructions. The fluorescence data were acquired using the fluorescence ratio PCR instrument (Roche, Shanghai, China). More than three biological and technical replicates were maintained. The relative gene expression was calculated using the 2^−∆∆CT^ method [27], and Ribosomal Protein S9 (*RPS9*) served as the reference gene [28,29].

### 2.10. Statistical Analysis

The data were statistically processed using analysis of variance (ANOVA) with Duncan’s multiple range (DMR) test in SPSS version 23.0 software (IBM SPSS Statistics, New York, NY, USA) [1] to analyze the differences in gene expression and fatty acid content. The data are expressed as mean ± SEM, and *p* < 0.05 is considered statistically significant.

## 3. Results

### 3.1. Analysis of Amino Acid and Protein Domain Conservation of Buffalo AGPAT Protein

By analyzing the AGPAT proteins of buffalo, cow, camel, goat, sheep, horse, pig, human, and mouse, the phylogeny of the *AGPAT* gene family was obtained (Figure 1A). In total, ten motifs were identified, and motif1 and motif2 were the most conserved (Figure 1B). Of the ten motifs, motif1, motif2, motif3, motif5, and motif6 were conserved in all proteins (Figure 1C). All of the AGPATs contain a PLN02380 superfamily, PLN02510 superfamily, or LPLAT-like domain (Figure 1C). Recent reports have confirmed that the aforementioned domain exhibited LPA acyltransferase activity, and used LPA as the substrate to catalyze acylation on the sn-2 site [30]. On analyzing the secondary structure of the AGPAT protein family, we observed that α and transmembrane helices are the main structures (Table 1). The tertiary structure analysis revealed that motif1, motif2, motif3, motif5, and motif6 of protein AGPAT1–5 formed a hydrophobic pocket (Figure 1D). The AGPAT6 tertiary structure exhibited a motif distribution significantly different from that of the other AGPAT proteins. qRT-PCR, performed for analyzing the expression of the buffalo *AGPAT* gene family during different lactation phases, revealed that *AGPAT3* and *AGPAT6* expressions were significantly higher in the middle and late lactation stages than in the earlier lactation stages (Figure 1D). We further analyzed the tertiary structure of the AGPAT family. Results showed that the tertiary structure of AGPAT1-5 is analogous except for one helix in the upper left corner, while that of AGPAT6 showed significant differences from other members (Figure 2A). The surface model further revealed that the four motifs form a “pocket” structure in AGPAT1-5 (Figure 2B). Electrostatic potential energy analysis revealed that this pocket is hydrophobic (Figure 2C). Further analysis of the domains showed that the phospholipid/glycerol acyltransferase domain was found in all of the AGPATs, suggesting that this “pocket” may be associated with the acylation function of the *AGPAT* gene in buffalo (Figure 2D). However, AGPAT6 showed a lysophosphatidylcholine acyltransferase LPCAT1-like domain, which is significantly different from other members (Figure 2D). The difference is consistent with the distribution difference in motifs found in the above spatial structure analysis. Based on the expression levels and tertiary structures of the AGPATs, we further explored the functions of *AGPAT1*, *AGPAT3*, and *AGPAT4* in the following study. 

### 3.2. Effect of AGPAT Interference on Gene Expression and Fat Synthesis in BMECs

The siRNA of *AGPAT1*, *AGPAT3*, and *AGPAT4* was transfected into the BMECs, and the transfected cells showed a typical cobblestone-like morphology (Figure 3A). The interference efficiency of each siRNA was analyzed, and the siRNA with the highest interference efficiency was then selected for the following experiments (Figure 3B). The transfection experiment was performed three times, and each sample was detected three times. The results show that after RNAi of *AGPAT1* was performed, the expressions of *GPAM* (glycerol-3-phosphate acyltransferase, mitochondrial), *PPARG* (peroxisome proliferator-activated receptor γ), *FASN* (fatty acid synthase), and *ACSL1* (acyl-CoA synthetase long chain family member 1) were significantly reduced (Figure 3C), and the total fatty acid content in the BMECs also reduced. After *AGPAT3* was interfered with, the expressions of *PPARG* and *ACSL1* were significantly reduced, whereas that of *LPIN1* (phosphatidate phosphatase LPIN1) was significantly increased (Figure 3D). The expression levels of *GPAM*, *DGAT2*, and *FASN* were significantly increased after *AGPAT4* interference, whereas the *LPIN1* expression level significantly decreased (Figure 3E). When *AGPAT1* and *AGPAT3* were interfered, the content of most types of fatty acids in the BMECs decreased significantly, whereas the content increased significantly after *AGPAT4* interference. Notably, the content of total fatty acids and UFAs in the BMECs decreased significantly after *AGPAT1* or *AGPAT3* interference (Table 2). However, after *AGPAT4* interference, the content of total fatty acids and monounsaturated fatty acids (MUFA) increased significantly, but no significant difference was observed in the total UFA content. Among them, the palmitic acid (C16:0) content significantly decreased after *AGPAT1* interference, whereas the content increased after *AGPAT4* interference. Regardless of whether *AGPAT1*, *AGPAT3*, or *AGPAT4* interference was performed, the content of α-linolenic acid (ALA, 18:3n-3), which is the precursor fatty acid of eicosapentaenoic acid (EPA, 20:5n-3) and DHA (docosahexaenoic acid, 22:6n-3), significantly decreased.

### 3.3. Effects of AGPATs Overexpression on Gene Expression and Fat Synthesis in BMECs

Following transfection of the *AGPAT* overexpression vector, green fluorescence was observed in the BMECs, indicating the successful expression of the overexpression vector in the cells (Figure 4A). The results show that at 48 h after the transfection, *AGPAT1*, *AGPAT3*, and *AGPAT4* expression in the BMECs increased by 38.33, 6.95, and 117.94 times, respectively, as revealed through qRT-PCR (Figure 4B). The expression of fat synthesis-related genes, including *ACSL1*, *DGAT2*, *FASN*, *GPAM*, and *PPARG*, significantly increased (Figure 4C–E). After *AGPAT1*, *AGPAT3*, and *AGPAT4* overexpression, the content of almost all types of fatty acids in the BMECs significantly increased, similar to the content of total fatty acids and total UFAs (Table 3). In addition, regardless of whether *AGPAT1*, *AGPAT3*, and *AGPAT4* were overexpressed, the content of ALA, arachidonic acid (ARA, 20:4n-6), EPA, and DHA significantly increased.

### 3.4. Potential Molecular Mechanism of AGPAT Gene Family Regulating Fat Synthesis in BMECs

Based on the aforementioned data, *AGPAT1* and *AGPAT3* expression were positively correlated with fat synthesis in the BMECs. *AGPAT1* or *AGPAT3* overexpression can increase the fat content in cells by promoting *FASN*, *PPARG*, *ACSL1*, *GPAM*, and *DGAT2* expression. Moreover, *AGPAT1* and *AGPAT3* expression had a greater effect on the UFA content than the SFA content, indicating that *AGPAT1* and *AGPAT3* play a key role in unsaturated fat synthesis in the cells. *AGPAT1* or *AGPAT4* interference led to increased expression of *ACACA*, which catalyzes the carboxylation of acetyl-CoA to malonyl-CoA. This is the first rate-limiting step of de novo fatty acid synthesis. This suggests that *AGPAT1* or *AGPAT4* is negatively correlated with de novo fatty acid synthesis, while *AGPAT1*, *AGPAT3*, or *AGPAT4* overexpression can promote the expression of *ACSL1*, which catalyzes the conversion of long-chain fatty acids (LCFAs) to their active form acyl-CoAs. Thus, *AGPAT1*, *AGPAT3*, or *AGPAT4* expression is positively correlated with long-chain fat synthesis in cells. In addition, medium- and long-chain fatty acids are potent ACACA inhibitors in the mammary gland, which further confirms the negative correlation between *AGPAT*s and de novo fatty acid synthesis. Surprisingly, *AGPAT4* overexpression and interference can both promote *DGAT2*, *FASN*, and *GPAM* gene expression and significantly increase the fatty acid content in cells. However, no rational explanation is available for this result at present. *AGPAT4* interference significantly decreased the PUFA content, whereas it significantly increased the SFA content. Simultaneously, *AGPAT4* overexpression significantly promoted PUFA synthesis, indicating that *AGPAT4* may exert a stronger catalytic effect on PUFA. Overall, *AGPAT1*, *AGPAT3*, and *AGPAT4* expression can all promote lipid synthesis in BMECs (Figure 5).

## 4. Discussion

The fat content and composition of milk are chief factors that affect milk flavor, nutritional value, and economic value. The *AGPAT* gene family plays crucial roles in milk fat synthesis [31,32]. However, the underlying regulatory mechanism in the buffalo has not been completely elucidated. Furthermore, current studies have mainly focused on the role of the *AGPAT* gene family in regulating milk fat content, with few investigating its role in regulating fat composition. In a previous study, the authors identified 32 and 14 AGPAT isoform protein sequences encoded by 13 *AGPAT* genes predicted from the river and swamp buffalo genomes, respectively [9]. However, most of the identified *AGPAT* genes are only predicted by sequence analysis, and cannot be amplified from the mRNA of MFG. One possible reason is that the *AGPAT* gene may have different specific splicing in different tissues. In this study, we amplified six *AGPAT*s from the MFG RNA of buffaloes and identified four conserved moieties in the amino acid sequences. Among them, the expressions of *AGPAT2* and *AGPAT5* are too low for further functional analysis. The functions of *AGPAT1* and *AGPAT6* in BMECs have been studied in a previous report [9,33]. We selected *AGPAT1* to compare this study with the previous report. Meanwhile, considering that AGPAT1 has a similar hydrophobic pocket with AGPAT3 and AGPAT4, we finally selected *AGPAT1*, *AGPAT3*, and *AGPAT4* for further study. RNAi and overexpression of *AGPAT1*, *AGPAT3*, and *AGPAT4* in the BMECs revealed that *AGPAT*s can promote fat synthesis in the BMECs by regulating the expression of fat synthesis-related genes, and had a stronger effect on UFA synthesis.

Milk fat synthesis is a highly coordinated process involving an extremely complex signal regulatory network. Diverse molecules participate in this process via multiple signal pathways. The main component of milk fat is TAG, and AGPAT catalyzes the conversion of LPA to PA, which is the second acylation step of TAG synthesis. AGPAT has been found to contribute to the diversity of glycerophospholipid species via selective esterification of fatty acyl chains at the sn-1 or sn-2 positions of membrane phospholipids. Many protein isoforms of the *AGPAT* gene family have been found, while the functions of different members may have a specialized role based on studies about genetic deficiencies in mice and/or humans. Based on sequence homologies, 11 LPAAT/AGPAT enzymes have been identified in mice and humans [34]. Among them, AGPAT4/LPAATδ is a physiologically essential enzyme that catalyzes the conversion of LPA (lysophosphatidic acid) to PA (phosphatidic acid), and is an essential component of the fission-inducing machinery driven by the protein BARS [34]. *AGPAT2* mutations can lead to congenital lipodystrophy [35], which indicates that other family members cannot replace the *AGPAT2* function. No significant differences in energy metabolism, food intake, and fat synthesis were observed in *AGPAT4* knockout mice, suggesting that the *AGPAT4* function may be compensated by other genes. In this study, both *AGPAT4* gene overexpression and RNAi promoted the expression of fat synthesis-related genes in the BMECs, and increased the cell fat content [11], which is consistent with the results of a previous study. In addition, each member of the AGPAT gene family was found to constitute a different evolutionary branch. We also identified ten conserved motifs through MEME screening, with five of them present in all *AGPAT* genes. However, only four of the motifs could form hydrophobic pocket structures, namely motif1 (EGTR), motif2 (NHXXXXD), motif3 (XXPXX), and motif6 (FXXR). Previous studies have found that the AGPAT family has four conserved motifs that are essential for substrate recognition and enzymatic activity: motif1 (xHxxxxD), motif2 (GxxFxxR), motif3 (xxEGxx), and motif4 (xxxxPxx) [30,36], which is consistent with our study. The report also pointed out that the four AGPAT motifs surround the putative acyl-CoA-binding pocket [30], and this pocket was also found in our results. Increasing the PUFA level is a vital method to improve the nutritional value of milk and dairy products. Different members of the *AGPAT* gene family have shown different activity on different fatty acid substrates. A previous study found that AGPAT3 prefers arachidonoyl-CoA, and AGPAT1 and AGPAT4 incorporate oleoyl-CoAs to lysophosphatidylethanolamine and lysophosphatidylserine [37]. In this study, we found that the arachidonate significantly increased after overexpression of *AGPAT3*, while oleic acid increased after overexpression of *AGPAT1* and *AGPAT4*, which is consistent with the report. 

Milk fat is obtained from two main routes; one route is by taking it directly from the blood, and the other route is through de novo synthesis in breast cells [38]. When lactation begins, the fat in milk is mainly taken from the blood, while during the second lactation week, the mammary cells begin to synthesize fat using acetic acid and butyric acid, with the synthesis reaching a peak on the 30th day of lactation [6]. Therefore, fat in milk from the second lactation month is mainly from the de novo synthesis of mammary cells. The fatty acid biosynthetic pathway is highly conserved, starting with acetyl-CoA carboxylation to malonyl-CoA, followed by malonyl-CoA condensation with acetyl-CoA to form LCFAs. In the mammary gland, fatty acids of C4–C14 and 50% of C16 fatty acids in milk fat are synthesized by acetic acid and β-hydroxybutyric acid. UFAs are mainly synthesized by the action of stearoyl-CoA desaturase (*SCD1*) to synthesize unsaturated fatty acids [39]. In addition, breast cells can also absorb LCFA from albumin–fatty acid-binding proteins (NEFA) and lipoproteins [6,40]. Triglycerides are further synthesized from fatty acids by enzymes such as GPAT, AGPAT, LPIN, and DGAT [41]. Among these enzymes, AGPAT catalyzes the conversion of LPA to phosphatidic acid, then further dephosphorylation to form DAG [42].

The efficiency of milk fat synthesis in breast tissue is not fixed at different lactation stages, and the expression patterns of *AGPAT* subtypes in breast cells are also different [11,42,43]. *AGPAT* is highly correlated with milk production traits such as milk fat percentage, milk fat composition, and milk fatty acid synthesis [31,32]. *AGPAT6* gene polymorphism can affect milk fat traits in dairy goats [44], and *ELOVL6* can regulate fat synthesis in BMECs by regulating *AGPAT6* expression [45]. *AGPAT1*, *AGPAT2*, *AGPAT3*, *AGPAT5*, and *AGPAT6* expression levels are significantly increased in pigs after 17 days of delivery, which indicates that *AGPAT* family genes are crucial for lactation in pigs [31]. In this study, the *AGPAT* gene family had a positive effect on fat synthesis in buffalo milk. This gene family can promote the de novo synthesis of fatty acids and TAG synthesis, especially UFA synthesis, by promoting *ACACA*, *FASN*, *ACSL1*, *GPAM*, *DGAT2*, and *PPARG* expression. *ACSL1* can promote milk fat synthesis and exhibits higher activity against the PUFA substrate [46]. In line with this, the study results also exhibited that *ACSL1* expression in buffalo was positively correlated with PUFA synthesis. However, in this study, *AGPAT4* overexpression or interference promoted *DGAT2*, *FASN*, and *GPAM* gene expression, and significantly increased the fatty acid content in cells. Currently, no rational explanation is available for this result, and further studies are warranted to reveal it.

## 5. Conclusions

In this study, we analyzed the amino acid sequences of the *AGPAT* gene family and identified 10 motifs in the proteins, among which motif1, motif2, motif3, and motif6 formed a hydrophobic pocket, which may be the functional domain in protein AGPAT1-5. Functional research showed that *AGPAT1* and *AGPAT3* plays a key role in the synthesis of fat, especially the UFA, in BMECs. However, further studies are required to reveal the function of *AGPAT4* during fat synthesis. This study provides new insights into different members of the *AGPAT* gene family on TAG synthesis in BMECs, and theoretical references for improving the fatty acid composition of milk.

## Figures and Tables

**Figure 1 genes-14-02072-f001:**
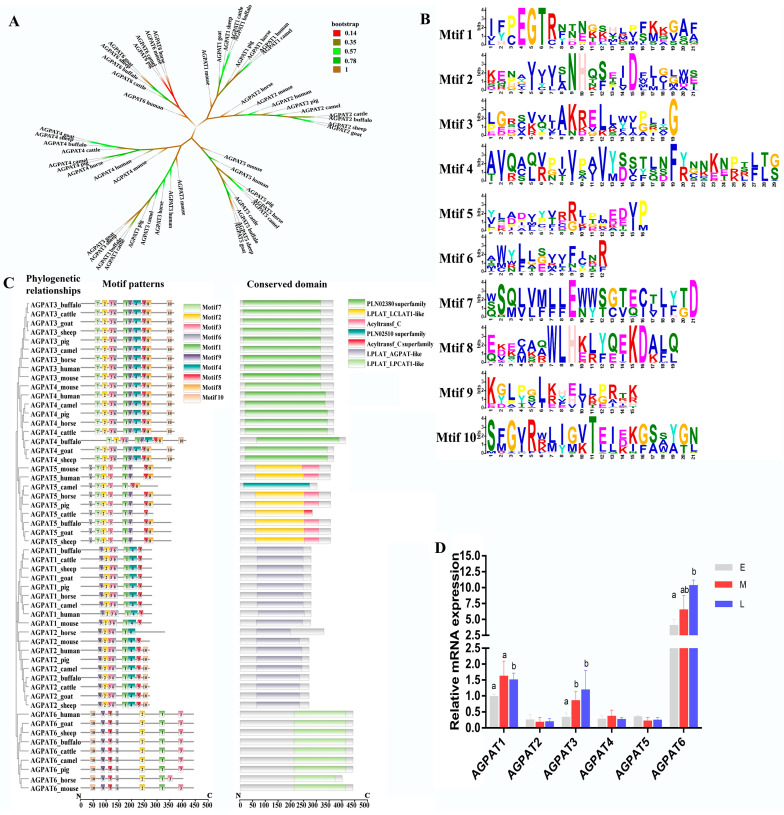
Sequences analysis of the *AGPAT* gene family. (**A**) Phylogenetic relationship of the *AGPAT* gene family. Different colors represent the credibility of the branches of the tree. (**B**) Motif patterns and conserved domains of AGPAT proteins. Larger amino acid letters represent greater conservation. (**C**) Motif and conserved domain analyses of the AGPAT family. (**D**) Expressions of *AGPAT*s in buffaloes at different lactation stages. Note: Different superscript letters indicate significantly differences (*p* < 0.05). E, M, and L indicate the early, middle, and late lactation stages, respectively.

**Figure 2 genes-14-02072-f002:**
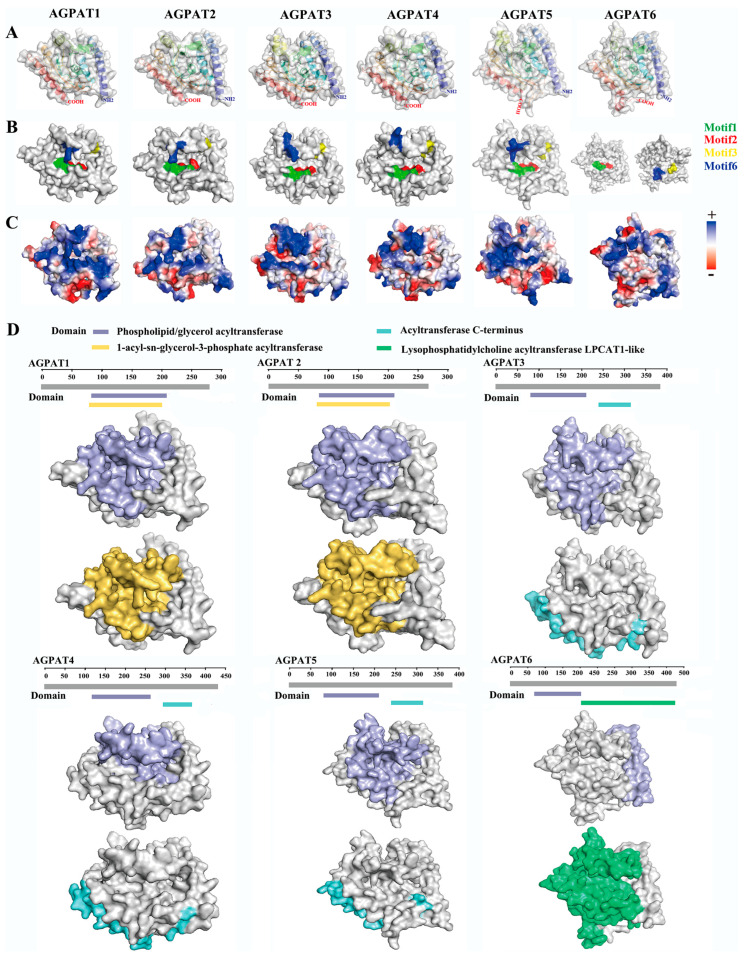
Tertiary structure, surface model, and electrostatic potential energy distribution of the AGPAT proteins in buffalo. (**A**) Tertiary structure of the buffalo AGPAT proteins. (**B**) The four motifs (motif1, motif2, motif3, motif6) in the surface model showed a pocket-like structure. (**C**) Electrostatic potential energy distribution of AGPAT proteins. Red and blue indicate negatively charged and positively charged regions, respectively. White indicates neutral/hydrophobic regions. (**D**) Domain analysis of the AGPAT proteins.

**Figure 3 genes-14-02072-f003:**
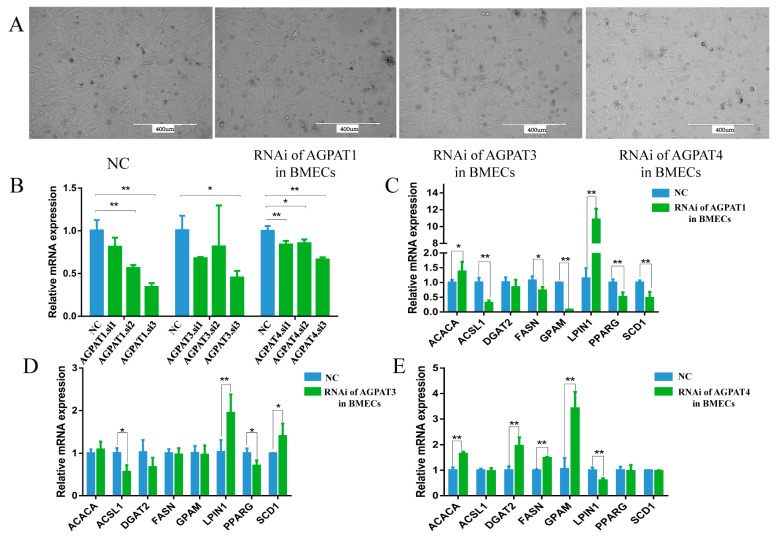
Effect of *AGPAT* interference on the expressions of fat synthesis-related genes in BMECs. (**A**) The morphology of BMECs after RNA interference. Scale bars = 400 μm. (**B**) The relative expressions of *AGPAT1*, *AGPAT3*, and *AGPAT4* after siRNA interference. (**C**) Effect of *AGPAT1* RNAi on the expressions of fat synthesis-related genes in BMECs. (**D**) Effect of *AGPAT3* RNAi on the expressions of fat synthesis-related genes in BMEC. (**E**) Effect of *AGPAT4* RNAi on the expressions of fat synthesis-related genes in BMECs. “*” represents *p* < 0.05; “**” represents *p* < 0.01.

**Figure 4 genes-14-02072-f004:**
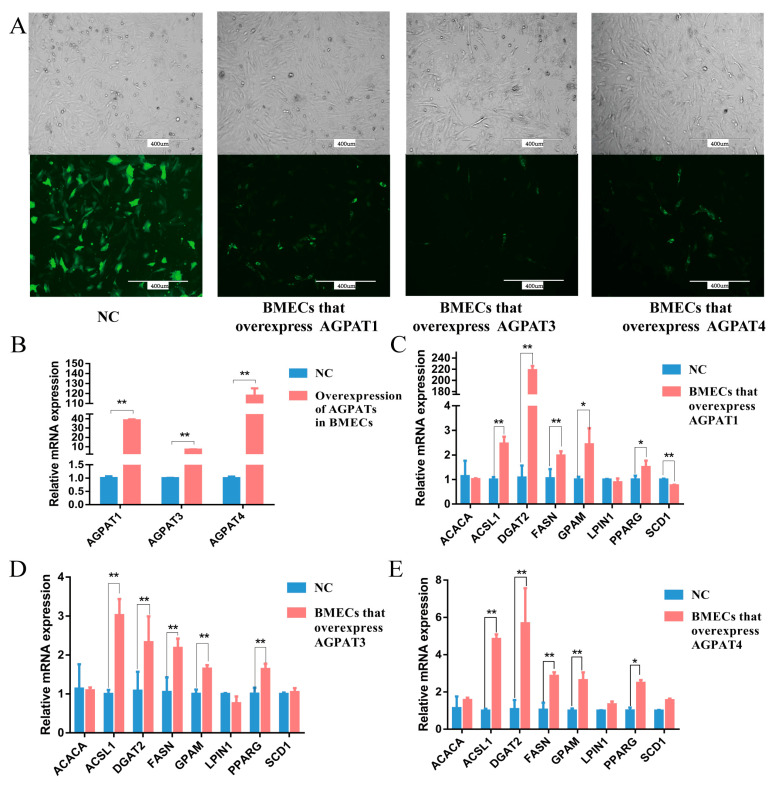
Effects of *AGPAT* overexpression on the expressions of fat synthesis-related genes in BMECs. (**A**) Transfection of the *AGPAT* overexpression vector in BMECs. Scale bars = 400 μm. (**B**) The relative expressions of *AGPAT1*, *AGPAT3*, and *AGPAT4* after the transfection. (**C**) Effects of *AGPAT1* overexpression on the expressions of fat synthesis-related genes in BMECs. (**D**) Effects of *AGPAT3* overexpression on the expressions of fat synthesis-related genes in BMECs. (**E**) Effects of *AGPAT4* overexpression on the expressions of fat synthesis-related genes in BMECs. “*” represents *p* < 0.05; “**” represents *p* < 0.01.

**Figure 5 genes-14-02072-f005:**
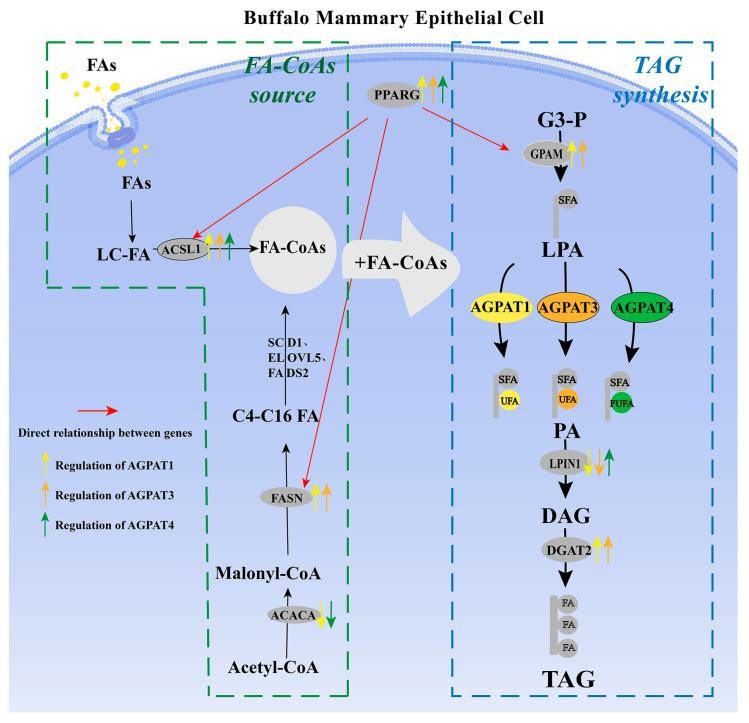
Effect of buffalo AGPAT on de novo synthesis of fatty acids and TAG synthesis. Note: FAs, fatty acids; LD, lipid droplet; and MFG, milk fat globules. The red arrow represents the change in gene expression with an increase in total fatty acid content. The black arrow represents the preferred fatty acid acyl group.

**Table 1 genes-14-02072-t001:** Secondary structure analysis of AGPAT proteins in buffalo.

	AGPAT1	AGPAT2	AGPAT3	AGPAT4	AGPAT5	AGPAT6
Number of amino acids	287	278	376	425	365	456
Disordered	13%	14%	12%	19%	11%	14%
α helix	52%	52%	56%	53%	56%	58%
β strand	14%	14%	12%	10%	14%	11%
TM helix	16%	17%	32%	29%	24%	29%

**Table 2 genes-14-02072-t002:** Changes in fatty acid content in BMECs after *AGPAT1*, *AGPAT3*, and *AGPAT4* were interfered.

Fatty Acid	NC	*AGPAT1* RNAi	*AGPAT3* RNAi	*AGPAT4* RNAi
C10:0	32.11 ± 1.08	12.43 ± 3.01 *	12.90 ± 2.49 *	15.03 ± 0.47 *
C14:0	44.97 ± 1.21	23.53 ± 0.58 *	26.53 ± 1.13 *	39.13 ± 2.96
C15:0	22.77 ± 0.59	13.23 ± 0.64 *	15.00 ± 0.45 *	20.53 ± 1.51
C16:0	120.40 ± 2.27	104.87 ± 2.30 *	119.27 ± 3.14	212.00 ± 25.27 *
C17:0	23.92 ± 0.50	16.70 ± 0.60 *	17.93 ± 0.22 *	24.27 ± 1.71
C18:0	89.91 ± 2.55	119.00 ± 5.69 *	122.03 ± 3.87 *	173.63 ± 3.55 *
C20:0	48.98 ± 1.79	29.93 ± 1.33 *	31.83 ± 0.55 *	41.50 ± 2.99 *
C22:0	52.44 ± 1.41	41.30 ± 1.42 *	42.50 ± 0.70 *	54.13 ± 5.06
C23:0	23.89 ± 0.57	16.67 ± 0.33 *	15.60 ± 1.17 *	20.23 ± 1.53
C16:1	24.36 ± 0.76	17.47 ± 0.47 *	19.97 ± 0.27 *	29.20 ± 4.13
C18:1n9c	90.58 ± 2.46	43.87 ± 1.56 *	44.60 ± 2.14 *	85.60 ± 5.70
C18:2n6c	30.86 ± 1.64	23.50 ± 0.38 *	27.23 ± 0.43	42.77 ± 3.47 *
C18:3n3	25.76 ± 0.35	9.33 ± 0.89 *	11.97 ± 1.51 *	16.93 ± 1.33 *
C20:1	24.81 ± 0.77	25.27 ± 0.67	26.93 ± 0.92	36.23 ± 2.18 *
C20:3n6	28.89 ± 1.11	18.50 ± 0.35 *	21.37 ± 0.63 *	24.47 ± 1.11 *
C20:3n3	26.00 ± 1.17	6.97 ± 1.52 *	7.50 ± 0.87 *	11.30 ± 2.12 *
C22:1n9	79.03 ± 3.05	121.03 ± 2.63 *	114.90 ± 2.77 *	155.07 ± 9.49 *
C20:4n6	30.55 ± 1.94	17.53 ± 2.22 *	21.77 ± 1.55 *	28.73 ± 3.24
C22:2	23.98 ± 0.57	19.20 ± 0.29 *	19.67 ± 1.41 *	27.90 ± 1.36
C20:5n3	26.07 ± 1.04	—	—	4.40 ± 0.35
C24:1	26.93 ± 0.82	30.63 ± 1.07	31.23 ± 0.69 *	42.67 ± 3.50 *
C22:6n3	22.53 ± 0.64	14.10 ± 1.75 *	20.10 ± 1.90	25.50 ± 3.48
LCFA	887.63 ± 12.78	713.97 ± 11.72 *	759.17 ± 4.39 *	1116.20 ± 25.56 *
Total SFA	459.39 ± 2.72	377.67 ± 10.27 *	403.60 ± 5.99	600.47 ± 11.98 *
Total MUFA	245.72 ± 5.67	238.27 ± 5.47	237.63 ± 1.87	348.77 ± 21.07 *
ω-3 PUFA	100.36 ± 3.06	31.73 ± 3.76 *	40.80 ± 1.64 *	58.13 ± 6.03 *
ω-6 PUFA	114.28 ± 4.48	78.73 ± 1.72 *	90.03 ± 2.19 *	123.87 ± 2.82
Total PUFA	214.64 ± 7.52	110.47 ± 5.36 *	130.83 ± 2.10 *	182.00 ± 8.63 *
Total UFA	460.35 ± 11.92	348.73 ± 2.91 *	368.47 ± 0.87 *	530.77 ± 29.34
Total FA	919.74 ± 13.86	764.50 ± 11.17 *	810.67 ± 6.73 *	1182.10 ± 27.78 *

Note: Data are shown as mean ± SEM. “*” indicates significant differences with the NC group. All fatty acid compositions are expressed as mg/100 g of fat. “—” indicates undetectable.

**Table 3 genes-14-02072-t003:** Changes in fatty acid content in BMECs after AGPAT1, AGPAT3, and AGPAT4 overexpression.

Fat Acid	NC	*AGPAT1*	*AGPAT3*	*AGPAT4*
C14:0	8.87 ± 0.75	33.40 ± 2.07 *	53.83 ± 1.98 *	33.40 ± 1.21 *
C15:0	4.20 ± 0.45	18.27 ± 1.36 *	28.10 ± 2.28 *	18.27 ± 0.82 *
C16:0	103.20 ± 8.59	133.90 ± 9.89	222.97 ± 36.02 *	116.80 ± 7.02
C17:0	8.47 ± 0.74	26.40 ± 1.54 *	39.47 ± 3.18 *	26.00 ± 0.95 *
C18:0	99.03 ± 11.17	131.63 ± 8.92	183.03 ± 6.68 *	121.87 ± 10.59
C20:0	17.60 ± 2.00	73.17 ± 4.65 *	107.13 ± 9.15 *	71.63 ± 2.93 *
C22:0	19.83 ± 2.09	83.40 ± 5.03 *	122.23 ± 10.13 *	81.33 ± 2.84 *
C23:0	10.40 ± 0.96	39.53 ± 2.20 *	59.00 ± 5.26 *	39.13 ± 1.56 *
C24:0	21.47 ± 2.34	90.90 ± 5.19 *	133.40 ± 11.62 *	88.20 ± 3.06 *
C16:1	5.77 ± 0.50	20.90 ± 1.27 *	31.43 ± 2.75 *	20.73 ± 0.92 *
C18:1n9c	32.50 ± 2.76	28.03 ± 2.85	42.27 ± 9.32	20.30 ± 4.59
C18:2n6c	22.20 ± 2.67	30.53 ± 2.97	42.37 ± 4.09 *	21.20 ± 1.24
C18:3n3	8.30 ± 0.87	33.10 ± 1.83 *	49.03 ± 4.37 *	33.50 ± 1.41 *
C20:1	11.27 ± 1.09	32.70 ± 1.94 *	48.20 ± 4.45 *	31.80 ± 1.26 *
C20:3n6	15.30 ± 1.78	35.63 ± 2.02 *	58.27 ± 4.74 *	37.37 ± 2.19 *
C20:3n3	9.47 ± 1.02	42.43 ± 2.37 *	63.33 ± 5.84 *	41.57 ± 2.12 *
C22:1n9	38.57 ± 3.97	120.60 ± 10.13 *	168.90 ± 18.05 *	113.27 ± 2.50 *
C20:4n6	14.17 ± 3.02	34.57 ± 1.89 *	52.93 ± 4.30 *	37.23 ± 1.34 *
C22:2	8.83 ± 1.11	40.57 ± 3.14 *	61.00 ± 6.42 *	39.83 ± 1.56 *
C20:5n3	6.50 ± 1.18	24.43 ± 0.88 *	36.17 ± 3.14 *	23.50 ± 0.86 *
C24:1	12.07 ± 1.18	42.90 ± 2.65 *	60.83 ± 5.28 *	40.87 ± 1.39 *
C22:6n3	7.70 ± 0.90	29.43 ± 2.58 *	45.47 ± 6.13 *	31.70 ± 2.31 *
LC-FA	485.70 ± 43.85	1146.43 ± 74.91 *	1709.37 ± 105.19 *	1089.50 ± 37.93 *
Total SFA	293.07 ± 28.80	655.53 ± 41.89 *	985.93 ± 38.59 *	620.83 ± 26.22 *
Total MUFA	100.17 ± 8.11	245.13 ± 17.11 *	351.63 ± 39.59 *	226.97 ± 4.19 *
ω-3	31.97 ± 3.71	129.40 ± 7.64 *	194.00 ± 19.35 *	130.27 ± 6.49 *
ω-6	60.50 ± 6.97	141.30 ± 10.01 *	214.57 ± 19.44 *	135.63 ± 3.91 *
Total PUFA	92.47 ± 10.61	270.70 ± 17.50 *	408.57 ± 38.77 *	265.90 ± 10.19 *
Total UFA	192.63 ± 18.60	515.83 ± 34.58 *	760.20 ± 78.28 *	492.87 ± 13.69 *
Total FA	507.60 ± 48.42	1171.37 ± 76.45 *	1748.80 ± 110.65 *	1113.70 ± 38.72 *

Note: Data are shown as mean ± SEM. “*” indicates significant differences with the NC group. All fatty acid compositions are expressed as mg/100 g of fat.

## Data Availability

The dates generated and analyzed during this study are included in this manuscript. Additional datasets used and/or analyzed during the current study are available from the corresponding author on reasonable request.

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
