# Peer review of "Role of Different Members of the AGPAT Gene Family in Milk Fat Synthesis in Bubalus bubalis"

_genes, 2023, doi:10.3390/genes14112072_

Round 1

Reviewer 1 Report

Comments and Suggestions for Authors

The manuscript presented for the review addresses an interesting and important issue concerning role of AGPAT gene family members in TAG synthesis and theoretical possibility of profitable modification of milk FAs composition.

My comments are as follows:

Introduction part - some information about buffalo milk fatty acids profile compared to other species should be included.

line 73 - method of mammarty gland tissue collection should be presented in bfrief

line 76 - method of milk collection should be presented briefly

line 146 - which "cells"? In what material were FAs analyzed? in line 162 authors mention milk

lines 213 - 217 - this is rather methodology part not results

lines 250 - 253 - as above

Refeences should be formatted according to journal guidelines

Author Response

Thanks for your comments concerning our manuscript. These comments are all valuable and very helpful for improving our manuscript, as well as the important guiding significance to our further researches. We have studied all comments carefully and revised our manuscript word by word. All of the suggestions have been revised and highlighted in the manuscript. We hope that this response and the revised manuscript will qualify enough to meet your requirement. The responses to the comments are listed in the following point-by-point.

  • Introduction part - some information about buffalo milk fatty acids profile compared to other species should be included.

R: Thanks for your suggestions. We have supplemented the information in the paper as follows. One of our previous studies showed that the milk fat in buffalo is significantly higher than that in Holstein milk (7.88±0.91 Vs 4.24±0.80). However, compared to Holstein milk, buffalo milk is rich in unsaturated fatty acids (UFAs), such as linoleic, linolenic, conjugated linoleic acid, eicosapentaenoic acid, and arachidonic acids, which are considered beneficial to human health.

  • line 73 - method of mammary gland tissue collection should be presented in brief

R: Thanks for your suggestions. The detailed method of mammary gland tissue collection was shown in the section “2.5 Isolation, culture, and purification of buffalo mammary epithelial cells”. Briefly, fresh buffalo mammary gland tissue was obtained from the butchery and washed three times with normal saline (0.9 % NaCl). The acinus portion was extracted from the mammary gland tissue, washed with normal saline (0.9 % NaCl), and transferred into high-resistance PBS (containing 400 IU mL-1 each of penicillin and streptomycin) until it was brought back to the laboratory.

  • line 76 - method of milk collection should be presented briefly

R: Thanks for your suggestions. We have supplemented the method of milk collection in the paper as follows. All of the selected buffalos were in the second or third parity with ages between 6.5 and 7 years. In brief, the milk samples were collected in summer (June–July), between 5:00–6:00 am, on the same day for each lactation stage. The samples were collected manually into sterile RNase-free tubes, taking care to avoid any contamination. The samples were immediately placed on ice and transported to the laboratory, and stored at −80 °C until further use.

  • line 146 - which "cells"? In what material were FAs analyzed? in line 162 authors mention milk

R: Thanks for your comments. We have revised the word “cells” to “BMECs transfected with siRNAs or overexpression vectors”. In line162, the material for FAs analyzed should be BMECs. We have revised it.

  • lines 213 - 217 - this is rather methodology part not results

R: Thanks for your comments. We have transferred the related description to the method part.

  • lines 250 - 253 - as above

R: Thanks for your comments. We have transferred the related description to the method part.

  • Refeences should be formatted according to journal guidelines

R: Thanks for your suggestion. We have revised the references based on the journal’s guidelines.

Reviewer 2 Report

Comments and Suggestions for Authors

·         Brief summary:

This investigation aims at undercovering the differences regarding the catalytic activity of different AGPAT proteins. The authors assessed conservation of amino acid sequences and protein domains of the AGPAT family, performing an analysis concerning the function of AGPAT1, AGPAT3, and AGPAT4 genes in buffalo mammary epithelial cells. Prediction of the AGPAT tertiary structure was performed and four conservated motifs were identified, including a hydrophobic pocket. This pocket was suggested as the active region of AGPAT enzyme.

·         General concept:

This paper did not establish an objective, which makes it difficult to identify testable hypotheses and the fulfillment of them across the reported results. The authors included analyses to detect conserved regions as well as domains in the AGPAT protein; however, most of the results only focused on the former conserved regions, which do not imply enzymatic activity (domains do). A test including multiple mammals was performed but in-depth analysis of these results was not achieved. Amongst these mammal species, neither human nor mice were included, which could have been used as reference given the higher degree of molecular research, including protein modeling and pathway analysis.

Protein modeling was used as a way to shed light on protein regions involved in the enzymatic activity of AGPAT; however, the methodology described seems insufficient. No information regarding modeling process and model quality is provided. A better description of the suggested enzymatic pocket should have been provided including a high-quality figure describing this structure. Protein modeling reproducibility by other research groups might be compromised.   

The analysis was focused on three members of the AGPAT family only. However, it is not clear why other members of this family were excluded. No exploration regarding isoforms was provided either. A previous report in buffalo suggests the existence of more reported proteins, which could have enriched this analysis. I cite the following: “32 and 14 AGPAT isoform protein sequences encoded by 13 AGPAT genes were predicted from the river and swamp buffalo genome, respectively.” A report made by Ma et al. (2022), which is found in the list of references but key results from this paper are not referenced.

Figures concerning cell cultures might be improved.

More bibliographic consideration is needed. The current draft does not only lack key results reported using buffalo as research model but also bibliography concerning better studied species such as human and mice.

Citation format is not adequate, e.i. formatting for reference 17 is different from reference 18, 19 and 20.

Author Response

Thanks for your comments concerning our manuscript. These comments are all valuable and very helpful for improving our manuscript, as well as the important guiding significance to our further researches. We have studied all comments carefully and revised our manuscript word by word. All of the suggestions have been revised and highlighted in the manuscript. We hope that this response and the revised manuscript will qualify enough to meet your requirement. The responses to the comments are listed in the following point-by-point.

General concept:

  • This paper did not establish an objective, which makes it difficult to identify testable hypotheses and the fulfillment of them across the reported results. The authors included analyses to detect conserved regions as well as domains in the AGPAT protein; however, most of the results only focused on the former conserved regions, which do not imply enzymatic activity (domains do). A test including multiple mammals was performed but in-depth analysis of these results was not achieved. Amongst these mammal species, neither human nor mice were included, which could have been used as reference given the higher degree of molecular research, including protein modeling and pathway analysis.

R: Thanks for your comments. The focus of this study was to investigate the mechanism through which different AGPATs on the efficiency of TAG synthesis and fatty acid composition. We have supplemented this information in the abstract and introduction section of the paper. The analysis of protein domains in this study was performed only to investigate the relationship between spatial structures and functional differences of the AGPAT proteins. The results showed that significant differences were found in motif pattern and conserved domain between AGPATs, indicating that protein structure may be one of the factors contributing to the differences in AGPAT enzyme activities. Maybe we should design a systematic and in-depth study specifically focusing on AGPAT enzymatic activity, but this was not the focus of this study. We have supplemented the human and mouse data in the phylogenetic and motif analysis. We also supplemented some human and mouse AGPAT-related researches in the discussion section.

  • Protein modeling was used as a way to shed light on protein regions involved in the enzymatic activity of AGPAT; however, the methodology described seems insufficient. No information regarding modeling process and model quality is provided. A better description of the suggested enzymatic pocket should have been provided including a high-quality figure describing this structure. Protein modeling reproducibility by other research groups might be compromised.

R: Thanks for your comments. We have supplemented the information regarding the modeling process and model quality, and a high-quality figure of the enzymatic pocket in the paper (Fig.2). The sequence of AGPAT was matched with the known structural sequence by Phyre2, and the remote homology detection method was used for 3D modeling. The models with confidence higher than 99% were selected for subsequent analysis.

  • The analysis was focused on three members of the AGPAT family only. However, it is not clear why other members of this family were excluded. No exploration regarding isoforms was provided either. A previous report in buffalo suggests the existence of more reported proteins, which could have enriched this analysis. I cite the following: “32 and 14 AGPAT isoform protein sequences encoded by 13 AGPAT genes were predicted from the river and swamp buffalo genome, respectively.” A report made by Ma et al. (2022), which is found in the list of references but key results from this paper are not referenced.

R: Thanks for your comments. The report you mentioned (Ma et al. 2022) is only a sequence-based prediction analysis of the genome, and the genome used in the study was provided by our group. In fact, by RT-PCR, we found that most of the predicted AGPATs can not be amplified. Since no product was amplified, we did not present this part of the results. One possible reason is that the AGPAT gene may have different specific splicing in different tissues. As shown in our study, we amplified 6 AGPATs from the MFG RNA of buffaloes. Among them, the expression of AGPAT2 and AGPAT5 are too low for further functional analysis. The function of AGPAT1 and AGPAT6 in BMECs has been studied in the report you mentioned (Ma et al. 2022). We selected AGPAT1 to compare this study with the previous report. Meanwhile, considering that AGPAT1 has a similar hydrophobic pocket with AGPAT3 and AGPAT4, we finally selected the three members of the AGPAT family only for further study.

  • Figures concerning cell cultures might be improved.

R: Thanks for your suggestion. We have tried our best to improve this figure in the paper. Probably due to the aging of our cell imaging equipment, this is the best picture we can get.

  • More bibliographic consideration is needed. The current draft does not only lack key results reported using buffalo as research model but also bibliography concerning better studied species such as human and mice.

R: Thanks for your suggestion. We have browsed the AGPAT related repots in the pub-med again to revise our study. Especially, we referred to a recent report which reviewed the biochemical and biological activities of the AGPATs, including their predicted structures, involvements in human diseases, and essential physiological roles as revealed by gene deficient mice (Valentine, William J, 2022). We have supplemented some essential analysis and references in the discussion section.

  • Citation format is not adequate, e.i. formatting for reference 17 is different from reference 18, 19 and 20.

R: Thanks for your suggestion. We have revised all of the references based on the journal’s guidelines.

Specific comments:

  • Line 2: include scientific name, Bubalus bubalis.

R: Thanks for your suggestion. We have replaced the word “buffalo” to “Bubalus bubalis”.

  • Line 23: how many AGPATs are described in buffalo” Why did you study only three members of this family.? Include scientific name.

R: Thanks for your comments. As we explained above, we selected the three members based on the expression level and tertiary structure of the AGPATs. We have supplemented the related information in the discussion section.

  • Line 26: AGPAT6 shows up. Why “suggesting that this is the functional domain of the AGPAT protein”?

R: Thanks for your comments. This description is based on the information from the references. A report pointed out that the four AGPAT motifs surround the putative acyl-CoA-binding pocket (Valentine, William J, 2022). We have deleted the related description in the paper for lack of evidence in this study.

  • Line 31: what do you mean by “theoretical references.”

R: The “theoretical references” mean all of the functional analyses on AGPATs are based on BMECs in vitro, which may be different when used in the in vivo models. We have removed the word “theoretical” for ease of understanding.

  • Line 18: research objective is missing in the abstract.

R: Thanks for your comments. We have supplemented the objective in the abstract.

  • Line 18: make a clearer statement of materials and methods in the abstract.

R: Thanks for your suggestion. We have supplemented the information about materials and methods in the abstract.

  • Line 32: do not include words present in the title as keywords. Sort them alphabetically.

R: Thanks for your suggestion. We have revised the keywords in the paper.

  • Line 38: change “nutritional value, and economic value.” To “, and nutritional and economic value.”

R: Thanks for your suggestion. We have revised this describe in the paper.

  • Line 42: change “linoleic acid, linolenic acid, and arachidonic acid” to “linoleic, linolenic, and arachidonic acids”

R: Thanks for your suggestion. We have revised this description in the paper.

  • Line 48: change “de novo” to “de novo

R: Thanks for your suggestion. We have revised it in the paper.

  • Line 53: change “AGPAT catalyzes” to “AGPATs catalyze”, you are considering multiple proteins.

R: Thanks for your suggestion. We have revised it in the paper.

  • Line 55: define specific tissues where these genes are highly expressed

R: Thanks for your suggestion. We have replaced the description “in buffalo and cattle milk” with “in buffalo and cattle mammary gland during lactation”.

  • Line 63: you have “poly polyunsaturated fatty acid (PUFA)”

R: Thanks for your suggestion. We have revised this mistake in the paper.

  • Line 64: consider changing “effects” to something like “the mechanism through which different AGPAT gene members…”

R: Thanks for your suggestion. We have revised this description in the paper.

  • Line 67-70: this seems a result/conclusion.

R: Thanks for your suggestion. We have revised this description in the paper.

  • Line 67: missing research objective of this paper

R: Thanks for your suggestion. We have supplemented the objective in the paper.

  • Line 76: change “early lactation (30–100 days), mid-lactation (100–200 days), and late (>200 days)” to “early (30–100 days), mid (100–200 days), and late (>200 days) lactation”

R: Thanks for your suggestion. We have revised this description in the paper.

  • Line 77: include a brief description of methods from Arora et al. (2019), so readers can comprehend your analysis without checking another document. Include what temperature was used for milk transportation.

R: Thanks for your suggestion. We have supplemented the brief description of the methods in the paper. In brief, the milk samples were collected in summer (June–July), between 5:00–6:00 am, on the same day for each lactation stage. The samples were collected manually into sterile RNase-free tubes, taking care to avoid any contamination. The samples were immediately placed on ice and transported to the laboratory, and stored at −80 °C until further use.

  • Line 94: “Contaminated genomic DNA” I think you meant “genomic DNA” since your goal is to get clean RNA

R: Thanks for your suggestion. We have revised this mistake in the paper.

  • Line 101: what do you mean by “(v11.0.11)”? include reference for MEGA11

R: Thanks for your suggestion. We have replaced this description with a reference.

  • Line 104: there are a number of appropriate citation papers available on tool for iTOL (https://itol.embl.de/upload.cgi).

R: Thanks for your suggestion. We have supplemented a citation in the paper.

  • Line 105: the same can be said about MEME-MEME.

R: Thanks for your suggestion. We have supplemented a citation in the paper.

  • Line 105: include citation for NCBI. For CD-Search tool:

R: Thanks for your suggestion. We have supplemented a citation in the paper.

  • Line 107: Citation for TB tools

R: Thanks for your suggestion. We have supplemented a citation in the paper.

  • Line 109: Phyre2 citation instead of server’s web site http://www.sbg.bio.ic.ac.uk/phyre2

R: Thanks for your suggestion. We have supplemented a citation in the paper.

  • Line 113: same issue for PyMOL

R: Thanks for your suggestion. We have supplemented a citation in the paper.

  • Line 114: change “PyMOL software (2.5.2)” to “PyMOL 2.5.2 software” and missing citation

R: Thanks for your suggestion. We have supplemented a citation in the paper.

  • Line 118: what did you use for washing?

R: Thanks for your question. In the study, normal saline (0.9 % NaCl) was used to wash the tissues. We have supplemented the information in the paper.

  • Line 123: commercial information about elements used to culture these cells

R: Thanks for your suggestion. We have supplemented the information in the paper. The cells were cultured with F12/DMEN (Gibco, Waltham, MA, USA) containing 20% serum (Gibco, Waltham, MA, USA) with a cell incubator (Thermo Fisher Scientific, Waltham, MA, USA).

  • Line 132: citation GenBank

R: Thanks for your suggestion. We have supplemented a citation in the paper.

  • Line 152: you have “(Agilent DB23, California, USA; 30 m × 0.32 mm i.d.; film thickness: 0.25 μm)”. You included specifications of the piece of equipment and commercial information. Separate both pieces of information

R: Thanks for your suggestion. We have revised it in the paper.

  • Line 157: (Supelco, Bellefonte, PA), USA?

R: Thanks for your comment. Here should be (Supelco, Bellefonte, PA, USA). We have revised it in the paper.

  • Line 159: “(Supelco)” include proper company name, “Supelco Analytical Products” maybe?

R: Thanks for your comment. We have revised this description in the paper.

  • Line 159: Matreya Inc.? or Matreya LLC

R: Thanks for your comment. By checking the company's official website, we found it should be Matreya LLC. We have revised it in the paper.

  • Line 161: Supelco

R: Thanks for your comment. We have replaced this word with (Supelco Analytical Products, Bellefonte, PA, USA).

  • Line 165: “3 times” to “three times”

R: Thanks for your comment. We have revised the spelling in the paper.

  • Line 168: cite Oligo 7.0 software.

R: Thanks for your suggestion. We have supplemented a citation in the paper.

  • Line 169: reference for the SYBR qPCR Master Mix you used

R: Thanks for your suggestion. We have supplemented a citation in the paper.

  • Line 169: I think you meant “Nanjing Vazyme Biotech Co”

R: Thanks for your suggestion. We have revised this mistake in the paper.

  • Line 171: is it “Roche Diagnostics”?

R: Thanks for your comment. Yes, the instrument is from Roche Diagnostics. We have revised this description in the paper.

  • Line 173: You could use something like this instead:

R: Thanks for your suggestion. We have supplemented a citation in the paper.

  • Line 173: include gene name “Ribosomal Protein S9

R: Thanks for your suggestion. We have supplemented the gene name in the paper.

  • Line 179: what are you testing here? Difference for expression? Faty acid content?

R: Thanks for your comments. We used the analysis of variance (ANOVA) with Duncan’s multiple range (DMR) test to analyze the differences in gene expression and fatty acid content. We have provided the related information in the paper.

  • Line 177: “SPSS 17.0” change to “SPSS version0”. include citation.

R: Thanks for your suggestion. We have revised it and supplemented the citation in the paper.

  • Line 177: include citations for the statistical analysis.

R: Thanks for your suggestion. We have supplemented a citation in the paper.

  • Line 182: values equal and below ten should not be written as number. I suggest using motive naming including the number, something like motif1.

R: Thanks for your suggestion. We have revised the related description in the paper.

  • Line 184-186: I believe it is better to focus most of your results on the domain analysis, which represents biological function of AGPAT. Motifs are important but they might not represent actual active regions of this protein, and you are looking for the active enzymatic region.

R: Thanks for your suggestion. We quite agree with you. We tried to analyze the domain of the protein to find the active enzymatic region. However, we are not specialized in domain analysis. We only found some publicly available tools to do simple analyses. In this study, to reveal the functions of different members, we focused our study on cellular experiments. If you can kindly provide us with information or tools to help us further analysis about the protein domain, we would be very grateful to you.

  • Line 185-187: how did you establish function for these domains?

R: Thanks for your question. These descriptions were speculated from a combination of domain analysis and literature reports. Since this result may involve over-speculation, we have removed the relevant description from the article.

  • Figure 1: This plot should be split into two so figure resolution could improve. For example, figure 1c labels cannot be read, but it applies to all of them. What is being represented in figure 1B? motives across species for a specific gene? Not clear.

R: Thanks for your suggestion. We have divided figure1 into two figures. The label of Figure 1c has been revised. Figure 1B shows the sequence conservation of the motifs. The abscissa is the amino acid sequence, and the ordinate represents conservation. Larger amino acid letters represent greater conservation.

  • Figure 1 legend: it needs a better description. A legend show stand by itself, meaning the reader should be able to understand what is being shown without reading corresponding segments of the paper. Figure 1B: neither X nor Y axes can be read. Figure 1C: why there is no high concordance between motifs and domains? Since domains are the ones having biological activity, I think more focus should be placed on analysis for domain instead of motifs. You used protein sequence, why are you using 5’ and 3’ to design molecule polarity on your x axis? Use adequate labels to show protein sequence directionality. Figure 1E: color legend location is quite confusing. Locate it on the left portion of this graph.

R: Thanks for your comments. We have revised the figure legends in the paper.

Figure 1. Sequences analysis of the AGPAT gene family. (A) Phylogenetic relationship of the AGPAT gene family. Different colors represent the credibility of the branches of the evolutionary tree. (B) Motif patterns and conserved domain of AGPAT proteins. Larger amino acid letters represent greater conservation. (C) Motif and conserved domain analyses of the AGPAT gene family. (D) Expression of AGPATs in buffaloes at different lactation stages. Note: Different superscript letters indicate significantly different (P < 0.05). E, M, and L indicate the early, middle, and late lactation stages, respectively.

  • Figure 1D: surface model is difficult to compare between genes of this family since they are being shown from different perspectives. Please, use the same perspective for all surface models, similar to the cartoon structure you are showing there. All of them should be presented using the same perspective (between members of the family as well as category -cartoon structure, surface model and electrostatic energy diagram-). If you want to show different sides of these models you need to create another figure, always showing the same perspective. Legend colors for domains and motifs are the same. It is not clear what is represented by category in your models, domains, or motifs?

R:Thanks for your question. We have revised this figure in the paper as Figure 2. All of the models were shown in the same perspective. Since the four motifs of AGPAT6 are distributed on different sides, two sub-figures were made to show the motifs. The legend colors have been defined in the paper.

Figure 2. Tertiary structure, surface model, and electrostatic potential energy distribution of the AGPAT proteins in buffalo. (A) Tertiary structure of the buffalo AGPAT proteins. (B) The four motifs (motif1, motif2, motif3, motif6) in the surface model showed a pocket-like structure. (C) Electrostatic potential energy distribution of AGPAT proteins. Red and blue indicate negatively charged and positively charged regions, respectively. White indicates neutral/hydrophobic regions.

  • Line 200: you kept working with AGPAT1, AGPAT3 and AGPAT4. However, I think it is not clear why you excluded other members of this family from your analysis. The information you provided in this paragraph does not seem well established to exclude other members of this family. You kept working with AGPAT4, but this gene shows low expression across lactation. You said “protein AGPAT1–5 formed a hydrophobic pocket” and “AGPAT3 and AGPAT6 expression was significantly higher in the middle and late lactation stages”. Expression of AGPAT4 is not significant across lactation. Are you using lactation time as a way to select protein sequences for further analysis? if so, what lactation period and why it is being used to select genes for further analysis?

R: Thanks for your question. As shown in our study, we amplified 6 AGPATs from the MFG RNA of buffaloes. Among them, the expression of AGPAT2 and AGPAT5 are too low for further functional analysis. The function of AGPAT1 and AGPAT6 in BMECs has been studied in the report you mentioned (Ma et al. 2022). Meanwhile, considering that AGPAT1 has a similar “pocket-like” structure to AGPAT3 and AGPAT4, we finally selected the three members of the AGPAT family in the subsequent study. We have supplemented this information in the discussion section.

  • Line 220: please include the name of these genes, GPAM, PPARG, FASN, and ACSL1. Not clear how you identified these genes? Did you include a bigger panel of genes to assess expression? Line 223: include LPIN1’s name.

R: Thanks for your suggestion. We have supplemented the full names of these genes. We selected these genes based on the published literatures, including the report (Ma et al. 2022) you mentioned above, one of our previous studies (Li et al. Fatty acid biosynthesis and transcriptional regulation of Stearoyl-CoA Desaturase 1 in buffalo milk, 2020, BMC genetics), and (M, B., JJ, L. 2008. Gene networks driving bovine milk fat synthesis during the lactation cycle. BMC Genomics). We selected genes considered to play essential roles during milk fat synthesis for testing.

  • Figure 2a: It is not clear what is being represented. Legend of the scale bar is not visible.

R: Thanks for your comment. Figure 2a (now revised as Figure 3a) illustrates the morphology of BMECs after RNA interference. BMECs are easily undergo reticular differentiation in vitro, which can affect their gene expression and function. The cells in our study maintained the cobblestone-like morphology after RNA interference, which was consistent with the control group, suggesting the detection of cellular gene expression was reliable.

  • Figure 2b: you could assign a name to each siRNAs you are using instead of using “three different siRNA of AGPATs”

R: Thanks for your suggestion. We have defined a name for each siRNA in the figure.

  • Line 461: Reference 25 needs to be replaced and updated.

R: Thanks for your suggestion. We have replaced and updated this reference.

Round 2

Reviewer 2 Report

Comments and Suggestions for Authors

Please consider including an analysis centered on domains additionally to the one concerning motives. You could use a server such as prosite https://prosite.expasy.org/

Additional characterization for the analyzed sequences can also be performed. You can use a number of tools present on expasy (https://www.expasy.org/search/Protein%20secondary%20structure%20prediction%20)

I think these type of analysis could improve your analysis.

Author Response

Thanks for your comments. The suggestions have been revised and highlighted in the manuscript. We hope that this response and the revised manuscript will qualify enough to meet your requirements. The responses to the comments are listed below.

Comments and Suggestions for Authors

Please consider including an analysis centered on domains additionally to the one concerning motives. You could use a server such as prosite https://prosite.expasy.org/

Additional characterization for the analyzed sequences can also be performed. You can use a number of tools present on expasy (https://www.expasy.org/search/Protein%20secondary%20structure%20prediction%20)

I think these type of analysis could improve your analysis.

R: Thanks for your comments. We tried to use your recommended website for further analysis of the domain. Unfortunately, the mapping of the sequences was poor on the website. Therefore, we used the tool from InterPro (https://www.ebi.ac.uk/interpro/)for domain analysis. Results showed that the phospholipid/glycerol acyltransferase domain was found in all AGPATs, suggesting that this "pocket" may be associated with the acylation function of AGPAT gene in buffalo. However, AGPAT6 showed a Lysophosphatidylcholine acyltransferase LPCAT1-like domain, which is significantly different from other members. The difference is consistent with the motif distribution difference found in the above spatial structure analysis (Fig.2D).